# Long-Term Service Performance of Hard-Grade Asphalt Concrete Base Pavement Based on Accelerated Loading Test of Full-Scale Structure

**Yang Wu [1], Xingye Zhou \*, Xudong Wang and Lingyan Shan**

National Observation and Research Station of Corrosion of Road Materials and Engineering Safety in Dadushe Beijing, Research Institute of Highway Ministry of Transport, Beijing 100088, China
\* Correspondence: xy.zhou@rioh.cn

**Abstract:** In order to evaluate the feasibility of using hard-grade asphalt concrete in China's highway construction, based on the accelerated loading test of the full-scale track of the Research Institute of Highway Ministry of Transport (RIOHTrack), the long-term service performance of hard-grade asphalt concrete base pavement was studied, and its actual service effect was evaluated. The study found that hard-grade asphalt concrete has higher strength, modulus and high-temperature stability. Although the low-temperature performance is generally not as good as that of asphalt, with a penetration grade (PG) of 60/80, it has equivalent or better cracking resistance in the area where the RIOHTrack is located. During the five-year observation period, the rutting deformation of the pavement structure showed an annual fluctuation increase. In comparison, the rutting resistance of a hard-grade asphalt concrete base structure can be increased by approximately 16%, and the deflection value is smaller, so the long-term performance of the anti-rutting effect and structural bearing capacity are better. The study shows that hard-grade asphalt concrete base pavement is suitable for highway construction in parts of China, where the climate is similar to that of the RIOHTrack location, or the temperature is higher or the latitude is lower.

**Keywords:** low-grade asphalt concrete; long-term service performance; full-scale structure; accelerated loading test; rutting; deflection; pavement cracking

## 1. Introduction

Asphalt with low penetration is suitable for the production of high-modulus asphalt concrete [1] due to its high hardness and viscosity, and is often used as the base for new asphalt pavements or as a structural reinforcement layer in maintenance and renovation projects. The penetration grade (PG) of this asphalt at 25 °C is generally 10 to 40. It is also called hard-grade asphalt. Hard asphalt with a PG of 10/20 or 15/25 is mostly used in Europe [2,3]. The most widely used in China is road petroleum asphalt, with a PG of 20/40 [4].

Hard-grade asphalt and mixtures have been studied previously in Europe, and are more widely used. High-modulus asphalt concrete prepared with hard-grade asphalt first appeared in France in the 1980s [5,6]. In order to reinforce the heavy traffic asphalt pavement structure, asphalt with a PG of 15 is used as the base or lower surface layer in an asphalt mixture, and the amount of asphalt is increased to the same level as the surface layer, which significantly improves its resistance to permanent deformation. The fatigue life can be extended by 30%. After this success, France has formulated corresponding technical specifications [7,8], which has made hard-grade asphalt widely popular in French road construction. Currently, 50% of the French asphalt pavement base and lower surface layers are applied with high-modulus asphalt concrete [3]. Due to the good road performance of hard-grade asphalt and mixtures, they have attracted the attention of many countries in the world. In Switzerland [3,9], Spain [3,10], the United Kingdom [11], Portugal [12],

Belgium [3], South Africa, Australia [13], the United States [14], Canada [3], China [15], etc., corresponding research and applications have also been carried out, which have greatly promoted the development of this technology.

Recently, researchers have carried out a lot of research work on the preparation of hard-grade asphalt binder. Lee [16] mixed base asphalt with high-boiling point petroleum and 4% styrene-butadiene rubber to prepare hard-grade asphalt, which not only improved the asphalt stiffness, but also ensured its ductility. Peng [17] prepared hard-grade asphalt based on crude oil and vacuum residue through propane deasphalting, deep distillation, mild oxidation and additive modification. The road performance and mechanical properties of the prepared asphalt mixture were better than those of an SBS-modified asphalt mixture, and Meng [18] also came to the same conclusion. Yang [19] mixed the base asphalt with buton rock asphalt to prepare hard-grade asphalt, and the prepared asphalt mixture showed good durability and fatigue performance.

For hard-grade asphalt mixtures, there are many corresponding studies and applications, mainly focusing on road performance research in the laboratories, and long-term performance evaluation based on outdoor field tests. Amjad et al. [20] studied the properties of hard asphalt and high-modulus asphalt mixtures. Compared with conventional asphalt mixtures, the fatigue damage resistance was increased by 9.3%, and the resilience modulus at 60 °C was increased by 63%. Espersson et al. [21] conducted a study on the effect of temperature on high-modulus asphalt mixtures, and the results show that a high modulus could still be maintained at high temperatures. When used as the base of airport pavement, it can not only reduce the thickness of the pavement, but it also improves the fatigue performance. Yang et al. [22] applied a dissipative energy method to evaluate the fatigue properties of high-modulus asphalt mixtures. Capitao et al. [23] and Perret et al. [24] found, through full-scale tests and long-term observations on test roads, that the high-temperature performance of high-modulus asphalt mixtures is significantly better than that of conventional pavement materials. The accelerated loading test carried out by Montanelli [25] showed that the rutting depth of hard-grade asphalt concrete is basically the same as that of SMA, and is obviously better than that of ordinary asphalt concrete. De-Backer [17] and Lee et al. [16] also came to similar conclusions via field tests. Liu et al. [26] found that the rutting depth of hard-grade asphalt concrete pavements is about one-third of that of ordinary asphalt pavements. Bankowski et al. [27] conducted a long-term performance comparison test using HVS, and the results show that the fatigue life of hard-grade asphalt concrete pavement is much higher than that of ordinary asphalt pavement. Diefenderfer et al. [28] paved a hard-grade asphalt pavement test road at three locations, and the results show that its resistance to wheel-induced damage was significantly better than that of a conventional asphalt pavement. Guo et al. [29] measured the dynamic response of the pavement structure using on-site embedded sensors. The results show that the lateral and longitudinal strains at the bottom of the asphalt layer of the high-modulus asphalt pavement structure were smaller than in other asphalt pavement structures. High-modulus asphalt mixtures can significantly improve pavement structural performance.

In China, the research and application of hard-grade asphalt and mixtures started relatively late, and currently, it mainly focuses on the laboratory test research of asphalt materials and asphalt mixtures [30,31]. In actual projects, Hebei, Guangdong, Guangxi, Inner Mongolia and other provinces and cities have only constructed some test sections with short lengths, and have not yet carried out large-scale engineering application [32,33]. There are two main reasons for this. On the one hand, hard-grade asphalt concrete has good road performance, while the long-term service performance verification of practical engineering in China has not been carried out, and the reliability and applicability of this technology are still doubtful. On the other hand, hard-grade asphalt concrete has been widely studied and applied in Europe, while the asphalt pavement structure in China generally uses semi-rigid subbase or base. China's natural environment, road construction materials, pavement design methods, etc., are quite different from those in European countries, so the mature practices of European countries cannot be directly applied in China.

Therefore, in order to evaluate whether hard-grade asphalt concrete is suitable for use in highway construction in China, this study relies on the accelerated loading test of full-scale pavement to obtain service performance data of the whole life cycle. The long-term service performance of hard-grade asphalt concrete base pavement is studied to evaluate its actual service effect, and to provide a certain reference and guidance for its promotion and application in China.

The main purpose of this study is to examine the long-term service performance of hard-grade asphalt concrete base pavement, to determine whether this material is suitable for use in road construction in China. A test has been conducted on a full-scale test track of the Research Institute of Highway Ministry of Transport (RIOHTrack). The hard-grade asphalt concrete base pavement structure was placed in the straight-line section of the track. During the test, heavy trucks were used for accelerated loading. Given that the pavement structure simulated by this test track has been in service for long enough, the long-term service performance of the hard-grade asphalt concrete base pavement can be studied based on indicators of the pavement structure, such as cracking, rutting, and deflection obtained in the accelerated loading test, and thereby its performance in practical use can be evaluated.

## 2. Materials and Methods

### 2.1. RIOHTrack Full-Scale Road Test Track

RIOHTrack, which this study relies on, is the first full-scale road test track in China's road engineering field. It is located in Beijing, where the average temperatures in the coldest month and in the hottest month are −4.6 °C and 25.8 °C, respectively. RIOHTrack is an oval closed curve composed of a straight line and a circular curve, laid out in a north–south direction and arranged symmetrically. The total length of the route is 2.039 km. According to the linear characteristics of the test road and the purpose of the test, it is divided into two test sections: the straight line section and the curve section. There are 19 pavement structures in the straight section, and the layout is shown in Figure 1. Various pavement structures paved by the RIOHTrack are very representative in China, basically covering more than 90% of the commonly used pavement structure types on the asphalt pavement of China's expressways. Moreover, each test section of the RIOHTrack was constructed with the same structure thickness, pavement material, and construction technology as the actual project. The width of each test section is 7.5 m, and two inner and outer lanes are set; each lane is 3.75 m wide, and the acceleration loading of the real vehicle is mainly aimed at the inner lane. The section length is 50–60 m, and the test size is also completely consistent with the actual engineering project. Therefore, these test structures can better reflect the real situation of actual engineering projects, and the performance observations carried out on this basis are also very representative [34–36].

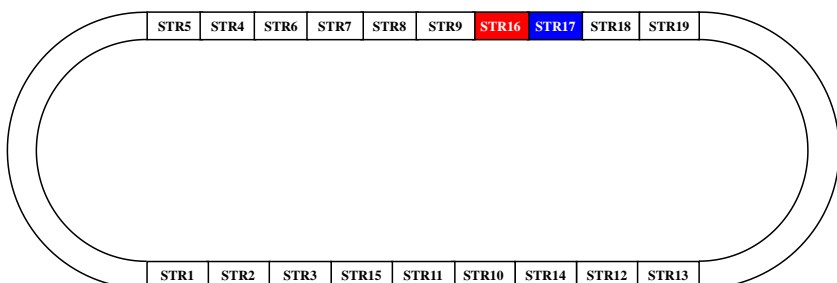

**Figure 1.** Pavement structure layout of RIOHTrack.

The hard-grade asphalt concrete base pavement structure in this study is STR17 on the straight test section, and the length is 60 m. The asphalt mixtures of the middle surface layer, the lower surface layer and the upper base layer all use AH-30 hard-grade asphalt with a PG of 20/40. The structure of the comparison road section is STR16, while the

asphalt mixtures of the lower surface layer and the upper base layer are conventional AH-70 asphalt with a PG of 60/80.

### 2.2. Structure and Materials

The pavement structure layout of STR17 and STR16 is shown in Figure 2. Here, the subbase of the STR17 is 20 cm cement-stabilized soil (CS), the lower base is 20 cm cement-bound gravel (CBG25), the upper base is 12 cm hard-grade asphalt concrete AC25 (AH-30), the lower surface layer is 12 cm hard-grade asphalt concrete AC25 (AH-30), the middle surface layer is 8 cm hard-grade asphalt concrete AC20 (AH-30), and the surface layer is 4 cm SMA13 (SBS). The thickness and material type of the subbase, lower base and surface layer of STR16 are the same as STR17; the difference is that the upper base and the lower surface layer are all 12 cm conventional asphalt concrete AC25 (AH-70), while the middle surface layer is 8 cm modified asphalt concrete AC20 (SBS).

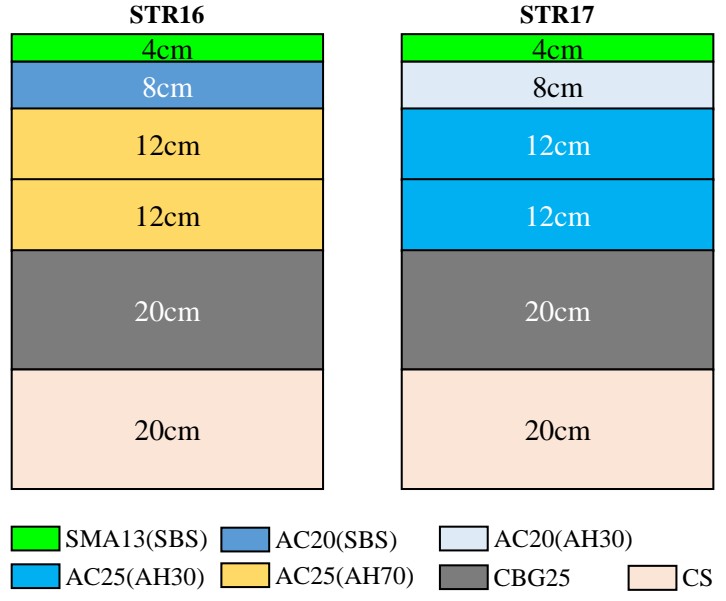

**Figure 2.** Pavement structure layout.

The pavement structures of STR16 and STR17 in this study have a total of five kinds of asphalt mixtures, including AC25 (AH-30), AC25 (AH-70), AC20 (AH-30), AC20 (SBS) and SMA13 (SBS). Among these, there are three kinds of asphalt, namely, AH-30, AH-70 and SBS. The basic performance test of asphalt is shown in Table 1. There are three kinds of asphalt mixture gradation, namely, SMA13, AC20 and AC25. The composition and gradation curve of mineral aggregate are shown in Table 2 and Figure 3. The volume parameters of the five asphalt mixtures are shown in Table 3, and the road performance tests are shown in Table 4 and Figure 4.

**Table 1.** Basic properties of asphalt.

| Index | Test Temperature | AH-30 | AH-70 | SBS |
|---|---|---|---|---|
| Ductility (cm) | 15 °C | 6.9 | >100.0 | - |
| | 10 °C | - | - | 49.4 |
| Penetration (0.1 mm) | 25 °C | 23 | 69 | 59 |
| Softening point (°C) | - | 61.1 | 49.1 | 77.8 |
| Rotary viscosity (Pa·s) | 135 °C | 1.362 | 0.496 | 2.718 |

**Table 2.** Mix design of HMA.

| Sieve Size (mm) | Percent Passing | | |
| --- | --- | --- | --- |
| | **AC25** | **AC20** | **SMA13** |
| 26.5 | 99.7 | - | - |
| 19 | 77.9 | 100 | - |
| 16 | - | 95.3 | 100.0 |
| 13.2 | 56.8 | 72.9 | 97.7 |
| 9.5 | 42.8 | 52.6 | 54.7 |
| 4.75 | 25.9 | 30.4 | 24.9 |
| 2.36 | 18.5 | 21.1 | 16.7 |
| 1.18 | 12.9 | 14.7 | 14.0 |
| 0.6 | 10.4 | 12.0 | 12.5 |
| 0.3 | 8.2 | 9.5 | 11.4 |
| 0.15 | 6.6 | 7.7 | 11.0 |
| 0.075 | 5.0 | 5.9 | 9.7 |

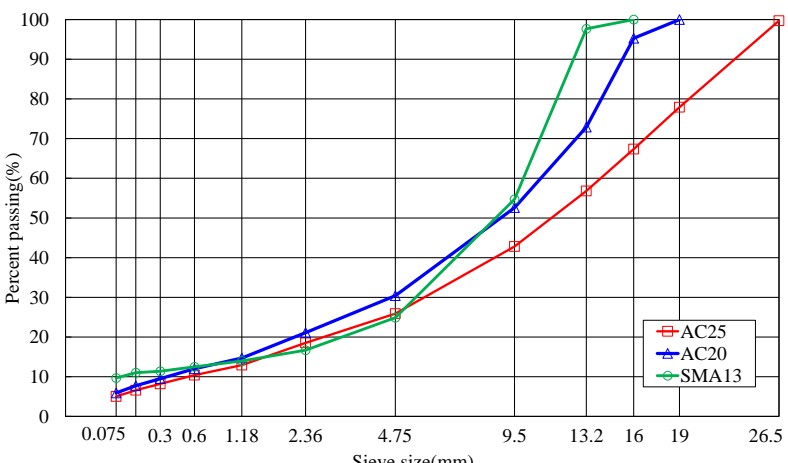

**Figure 3.** Asphalt mixture mineral aggregate gradation curve.

**Table 3.** Results of Marshall test of HMA.

| Index | AC25 (AH-30) | AC25 (AH-70) | AC20 (AH-30) | AC20 (SBS) | SMA13 (SBS) |
| --- | --- | --- | --- | --- | --- |
| OAC (%) | 4.12 | 4.07 | 4.60 | 4.48 | 5.52 |
| Bulk specific gravity | 2.537 | 2.531 | 2.541 | 2.541 | 2.470 |
| VV (%) | 3.8 | 4.1 | 3.4 | 2.9 | 4.5 |
| VFA (%) | 71.4 | 68.6 | 73.5 | 76.4 | 73.4 |
| VMA (%) | 13.3 | 13.4 | 13.3 | 13.2 | 17.3 |

**Table 4.** Performance of HMA.

| Performance Test | Index | Test Temperature | AC25 (AH-30) | AC25 (AH-70) | AC20 (AH-30) | AC20 (SBS) | SMA13 (SBS) |
|---|---|---|---|---|---|---|---|
| High-temperature rutting test | Dynamic stability (times/mm) | 60 °C | 6869 | 2287 | 7144 | 7952 | - |
| | | 70 °C | - | - | - | - | 4081 |
| Low-temperature beam bending test | Maximum tensile strain at failure (με) | 0 °C | 1555 | 2304 | 1628 | 3570 | 6254 |
| | | −10 °C | 1121 | 1312 | 1358 | 1817 | 2320 |
| | | −20 °C | 1040 | 1382 | 1114 | 1464 | 1691 |
| | | −30 °C | 875 | 1224 | 1143 | 1146 | 1394 |
| Modulus test | Dynamic modulus (MPa) | 20 °C | 19,189 | 12,004 | 17,486 | 10,528 | 7307 |

Cement-stabilized materials are of two types: cement-stabilized soil (CS) and cement-bound gravel (CBG25). CS uses cement P.O32.5, and the stabilized material is the undisturbed soil where the RIOHTrack is located. In the mix design, the optimum cement content is 9.5%, the optimum moisture content is 15.5%, the maximum dry density is 1.816 g/cm$^3$, and the 7-day unconfined compressive strength should not be less than 2 MPa. CBG25 uses cement P.O32.5, and the stabilized material is crushed limestone. In the mix design, the optimum cement content is 6%, the optimum moisture content is 5.59%, the maximum dry density is 2.434 g/cm$^3$, and the 7-day unconfined compressive strength should not be less than 6 MPa.

*2.3. Accelerated Loading Test Methods*

This study is mainly conducted on the RIOHTrack, which was completed in October 2015, and has been officially in operation for loading test since December 2016. To date, the RIOHTrack has been in operation for more than five years for accelerated loading tests using real vehicles.

In order to ensure the test efficiency, RIOHTrack is loaded with a heavy truck. From December 2016 to December 2018, loading mode I was used for loading, with four three-axle trucks. The axle load of the front axle of each vehicle was 8 tons, the axle load of the single rear axle was 16 tons, and the driving speed was 55 km/h, as shown in Figure 5. From January 2019, loading mode II was adopted for loading, and the loading vehicles were upgraded to six-axle trucks, as shown in Figure 6, with a total of six vehicles. The test loading efficiency was improved more than 3 times compared to that of loading mode I, and the driving speed was 45 km/h. Among them, the axle loads of the vehicles in loading mode II in winter and spring were as follows: the axle load of the front axle was 7 tons, the axle loads of the two middle axles were 14 tons, and the axle loads of each of the three rear axles were 18 tons. The axle loads in the summer and autumn were 7 tons for the front axle, 14 tons for the middle two axles, and 16 tons for each of the three rear axles.

Due to the heavy axle load of the loaded vehicle, which exceeded the standard axle load of 10 tons stipulated in China's Specifications for Design of Highway Asphalt Pavement, it had to be converted into ESALs under the action of standard axle load, according to Equation (1) [37]. This is used to characterize the traffic load level that the pavement structure bears, and it corresponds to the design life of the pavement structure.

$$N = C_1 C_2 \left( \frac{P_i}{100} \right)^{4.35} \tag{1}$$

where

$C_1$—axle number coefficient of the converted vehicle;
$C$—wheel set coefficient of the converted vehicle;
$P_i$—axle load of the converted vehicle (t).

According to this calculation, the cumulative number of ESALs in the RIOHTrack loading test reached 60.44 million from December 2016 to February 2022, which is equivalent to more than the traffic load level of the expressway heavy traffic level in China's Specification for the last 30 years. The cumulative ESALs curve is shown in Figure 7.

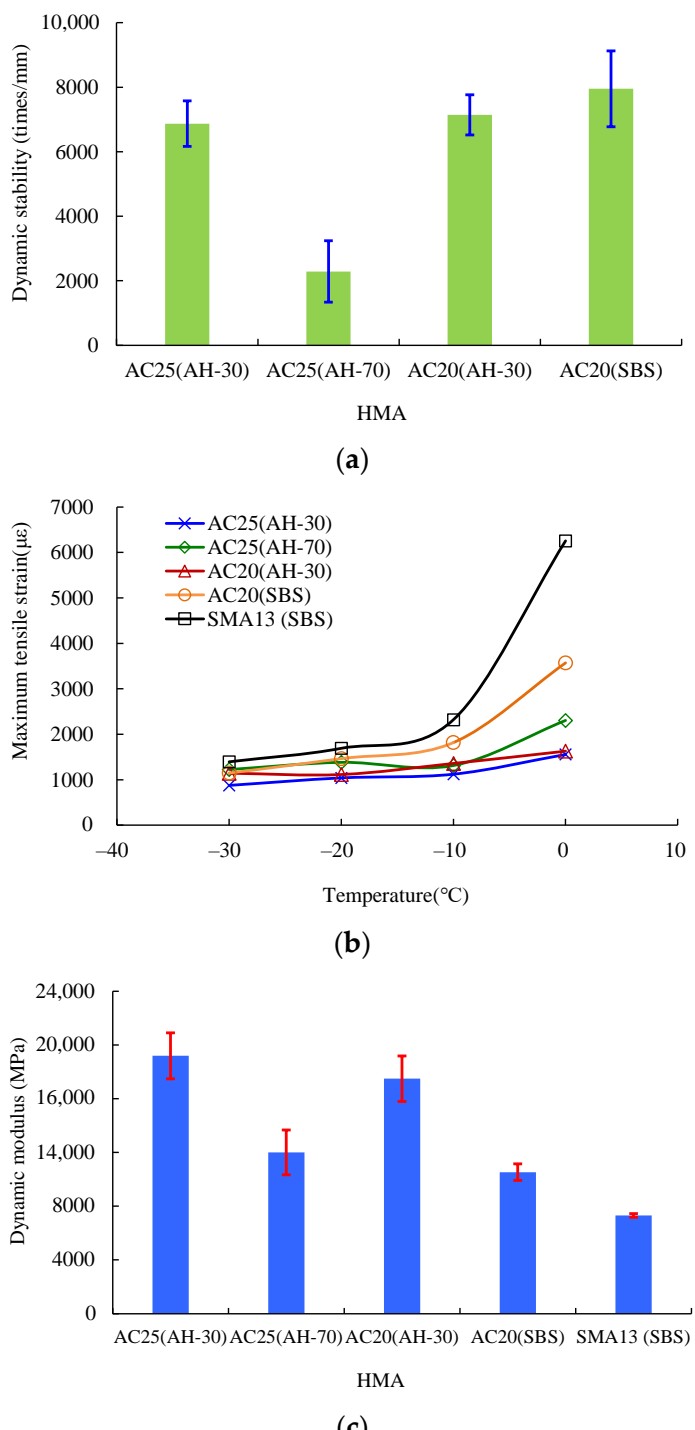

**Figure 4.** Asphalt mixture road performance test: (**a**) high-temperature performance test of asphalt mixture; (**b**) low-temperature performance test of asphalt mixture; (**c**) asphalt mixture dynamic modulus test.

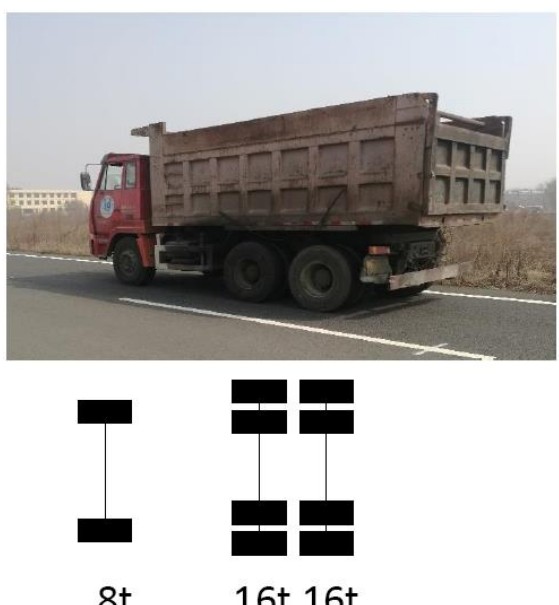

**Figure 5.** Distribution of axle load and axle shape of the truck in loading mode I.

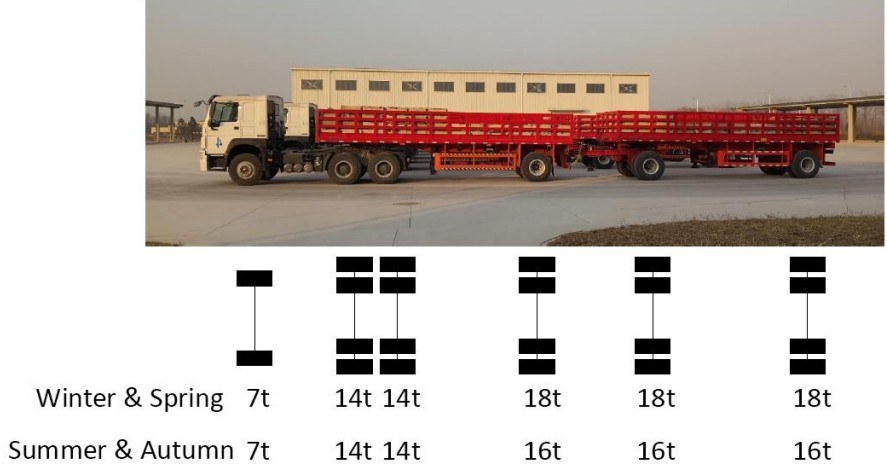

**Figure 6.** Distribution of axle load and axle shape of truck in loading mode II.

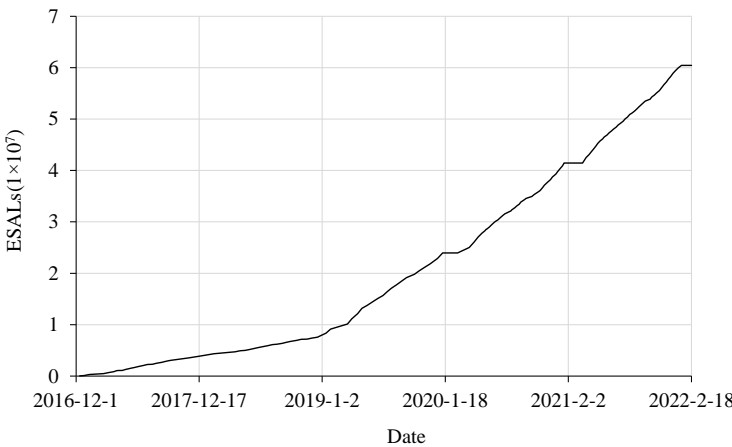

**Figure 7.** Curve of ESALs.

In order to study the service performance and actual response of the pavement, 23 sensor monitoring sections and more than 2000 sensors were laid in the asphalt pavement test section, which could realize the real-time monitoring of the pavement's service state. During the loading operation of the track, periodic tests were carried out two to four times a month to test the service performance in relation to disease, bending, rutting, flatness, skid resistance and other test sections. Up to now, 115 test cycles of data collection have been completed. As mentioned above, the accelerated loading time of RIOHTrack is equivalent to the 30-year level of the expressway heavy traffic level, so we could obtain indicators of the pavement structure's cracking, rutting, deflection and other things during the service process to study the long-term service performance of hard-grade asphalt concrete base pavement, and evaluate its actual service effect.

## 3. Long-Term Service Performance of Pavement Structure

### 3.1. Cracking

Biweekly examinations of pavement cracks are conducted as part of the observation of long-term performance. As of September 2021, the cumulative number of ESALs of the RIOHTrack loading test had reached 52.71 million. For both types of pavement structures, no transverse cracking, longitudinal cracking, or map cracking has been observed in road sections, other than transverse cracks (Figure 8) at the corresponding positions on the road surface where the sensors are embedded in the structure. As the data collection cable of the sensor embedded in the structure needed to be wrapped in a PVC tube and stretched horizontally from 12 cm under the surface inside the structure to the collection box on the road side, reflection cracking tended to appear at the corresponding position on the road surface where the PVC tube was placed beneath. However, such cracking cannot directly reflect the performance difference between the pavement material and the structure, but rather reflects an internal defect in the structure.

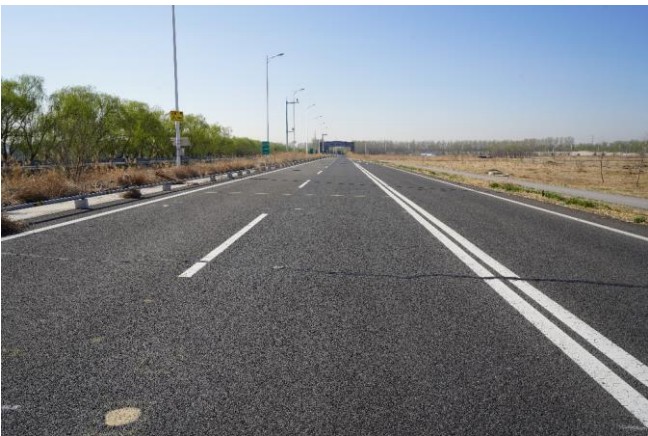

**Figure 8.** Transverse cracking where the sensor is embedded.

To more accurately evaluate the internal damage in the two pavement structures, 3D radar was used to detect the internal defects of the two pavement structures in September 2021. Table 5 shows the distribution of cracking obtained by radar detection, on the top surface of the upper base AC25 (AH-70) of STR16 and in the upper base AC25 (AH-30) of STR17, except where the sensors were embedded. Table 5 shows that four transverse cracks were detected in the asphalt concrete bases of both structures.

**Table 5.** Statistical table of cracks.

| Structure Number | Crack Stake | Crack Type | Crack Length (m) | Total Crack Length (m) |
|---|---|---|---|---|
| STR16 | K1 + 073.600 | transverse crack | 4.6 | 26.3 |
| | K1 + 105.100 | transverse crack | 7.1 | |
| | K1 + 110.000 | transverse crack | 7.2 | |
| | K1 + 122.400 | transverse crack | 7.4 | |
| STR17 | K1 + 138.600 | transverse crack | 7.1 | 22.9 |
| | K1 + 146.700 | transverse crack | 3.7 | |
| | K1 + 166.500 | transverse crack | 6.8 | |
| | K1 + 176.140 | transverse crack | 5.3 | |

In STR16, three out of the four transverse cracks were found to run across both the inner lane and the outer lane (K1 + 105.100, K1 + 110.000, K1 + 122.400), and only one did not run all the way through. The total length of all the cracks was 26.3 m. In STR17, one out of the four transverse cracks was found to run across both the inner lane and the outer lane (K1 + 138.600), and three of them did not run through. The total length of all cracks was 22.9 m. We know that transverse cracks generally include temperature cracks and load-type cracks. Regarding the cracks of STR16 and STR17, for the transverse through-type cracks whereby both inner and outer lanes crack, temperature cracks were the main ones. This is because, during the RIOHTrack test, only the inner lane was loaded, and the outer lane was not loaded. When transverse cracking occurred in the outer lane, especially a through-type crack in which both the inner and outer lanes cracked, this could only have been a shrinkage crack caused by temperature change, not a load-type crack caused by load. It can be seen from Table 5 that three through-type cracks appeared on the top surface of the base layer of STR16, and only one through-type crack appeared on the top surface of the base layer of STR17. The total cracking length of STR16 was 3.4 m longer than that of STR17 (14.8% longer). This shows that the base layer AC25 (AH-70) of STR16 was less resistant to temperature stress than the base layer AC25 (AH-30) of STR17. That is, the pavement structure STR17, using hard-grade asphalt concrete as the base, had better resistance to structure cracking than the pavement structure STR16, which used AH-70 asphalt concrete as the base. Combined with the temperature field inside the pavement structure, this study further analyzed the differences in the low-temperature cracking resistances of two asphalt concrete bases, STR16 and STR17. To obtain comprehensive temperature field data, we embedded temperature sensors inside the pavement structure (Figure 9). The equipment type was a PT100 platinum resistance temperature sensor, with the measurement range of −50 to 100 °C and a resolution of 0.1 °C. The embedding position of the temperature sensor was in each bottom layer of the pavement structure, and inside the subgrade 150 cm, 200 cm and 250 cm from the pavement surface. The data were collected every 10 min, and the collection method was 24 h continuous collection. Based on the results from the temperature sensor embedded in the pavement structure, from 2017 to 2021, the annual minimum temperatures of the two pavement structures at the positions of 12, 24, and 36 cm beneath the road surface are shown in Table 6 and Figure 10.

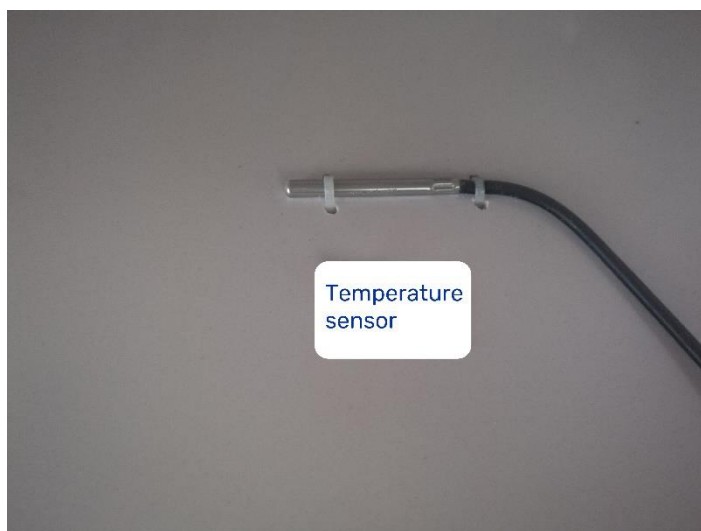

**Figure 9.** Temperature sensor.

**Table 6.** Statistical table of annual extreme minimum temperatures inside the structure.

| Years | STR16 (°C) | | | STR17 (°C) | | |
|---|---|---|---|---|---|---|
| | h = 12 cm | h = 24 cm | h = 36 cm | h = 12 cm | h = 24 cm | h = 36 cm |
| 2017 | −5.6 | −4.1 | −2.9 | −7.1 | −4.7 | −3.4 |
| 2018 | −8.7 | −6.8 | −5.2 | −9.0 | −6.6 | −5.2 |
| 2019 | −8.0 | −6.5 | −5.2 | −8.4 | −6.5 | −5.1 |
| 2020 | −8.4 | −6.5 | −4.8 | −8.6 | −6.1 | −4.4 |
| 2021 | −12 | −9.4 | −7.3 | −12.6 | −9.5 | −7.2 |
| Mean value | −8.5 | −6.7 | −5.1 | −9.2 | −6.7 | −5.1 |

　　　The results indicate that within the structure STR17, the most unfavorable temperature range for the service of the hard-grade asphalt concrete base was −5 °C to −9 °C, which falls in the range of 0 °C to −10 °C. According to the low-temperature performance results in Table 4, within the temperature range of 0 °C to −10 °C, the ultimate failure strain of hard-grade asphalt concrete AC25 (AH-30) is 1121–1555 µε, while in the case of asphalt concrete AC25 (AH-70), it is 1312–2304 µε, indicating a certain difference in the low-temperature crack resistance between the two. When the temperature was lower, the low-temperature crack resistance of hard-grade asphalt concrete became much worse. This contradicts the actual observation results shown in Table 5. Since the test results in Table 4 show relative performance evaluations that simplify the test conditions in the laboratory, the data in Table 5 are the actual observations of the pavement after accelerated loading tests. Given that the latter is closer to the practical engineering condition, it is more accurate to use the actual observation results when evaluating pavement cracking. From the previous actual observation results, it can be seen that, compared with STR17, STR16 had two more through-type cracks, the total length of cracking was 3.4 m longer, and the percentage of cracking was 14.8% higher. Based on this, we believe that, for areas where RIOHTrack is located, using hard-grade asphalt concrete as the base of the pavement structure can guarantee a crack resistance equivalent to or slightly better than that of the AH-70 asphalt concrete. Thus, this type of asphalt concrete can be used in practical engineering projects, and also has certain applicability in areas with a climate similar to that at RIOHTrack, with either a higher temperature or a lower latitude.

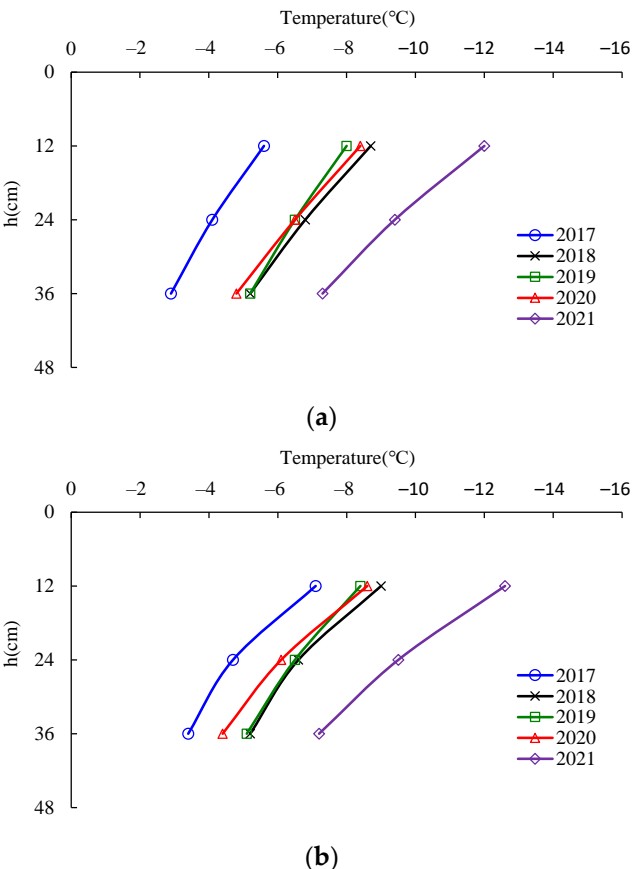

(a)

(b)

**Figure 10.** Annual extreme minimum temperature distribution inside the pavement structure: (**a**) STR16; (**b**) STR17.

*3.2. Rutting*

To obtain the long-term rutting curve, from December 2016 to April 2020, biweekly road rutting detection was conducted, and from May 2020 to the present, weekly road rutting detection has been conducted. Figure 11 shows the rutting curves and ESALs for the two pavement structures from December 2016 to February 2022.

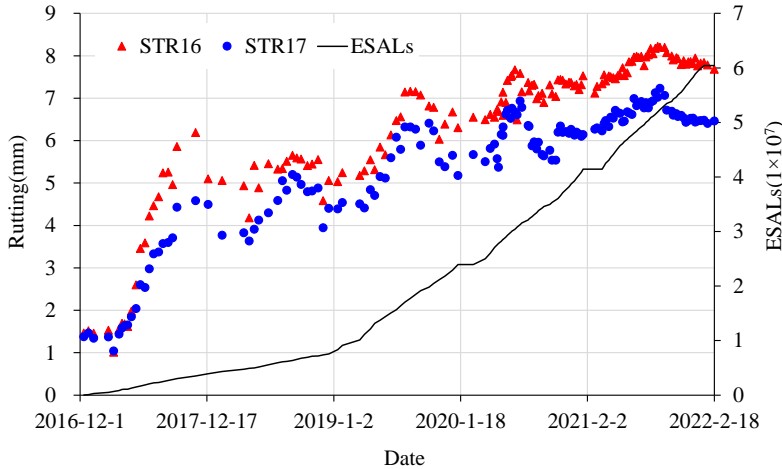

**Figure 11.** Evolution curve of rutting and accumulated axle times.

Figure 11 shows only a slight difference in rutting between the two pavement structures in the initial stage of loading, but with the increase in the cumulative number of axle loads, the difference between the two emerges, and gradually becomes significant.

From the rutting test results in Table 4, we can observe that at high temperatures, the performance of AC20 (SBS) asphalt concrete in the lower surface layer of STR16 was 11.3% better than that of AC20 (AH-30) in the same layer of STR17. Therefore, the difference in rutting between the two structures is mainly due to the material of the asphalt concrete base, and the overall rutting in structure STR17, which uses hard-grade asphalt concrete as the base layer, was less serious.

Figure 12 shows the rutting curves of the two pavement structures and the average temperature change from December 2016 to February 2022. Figure 12 shows that rutting has undergone an annual fluctuating increase.

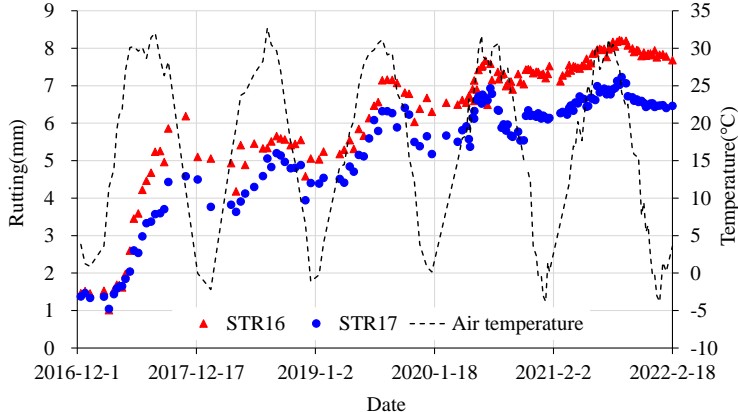

**Figure 12.** Evolution curve of rutting and air temperature.

In the five-year observation period, although the amount of rutting generally increased with the increase in the number of loads, fluctuations following a one-year cycle can also be observed, that is, when the temperature was high in summer, the amount of rutting reached the maximum value for the year, and then with the decrease in temperature, the rutting began to recover, until the amount reached the minimum value for the year at the lowest temperature in winter. Within the one-year cycle, rutting recovered to a certain extent, which is a different finding from the existing assumption that rutting is a plastic irrecoverable deformation [38]. No scientific explanation for this phenomenon is available at present. The US National Center for Asphalt Technology also discovered a similar phenomenon in their rutting observations of some test sections on a full-scale test track [39]. Therefore, as speculated, this phenomenon occurs when the amount of rutting increases. While annual fluctuations might be a pattern that has long existed, it can only be discovered when rutting detection is conducted more frequently, and the amount of rutting is recorded in all 12 months of the year.

To quantitatively characterize the difference in rutting between the two structures, the difference in the amount of rutting between the two structures detected each time was calculated. The result shows that the difference in the amount of rutting between STR16 and STR17 is in the range of 0.04–1.82 mm, with an average value of approximately 1 mm. In February 2022, the rut depth of STR16 was 7.68 mm, and that of STR17 was 6.46 mm. The rutting resistance of STR17 was approximately 16% higher than that of STR16, which means that the rutting resistance of the hard-grade asphalt concrete base pavement structure was better. This is mainly due to the higher hardness and viscosity of hard-grade asphalt, and the better rutting resistance of the asphalt mixture. When this material was used in base pavement, it greatly improved the high-temperature deformation resistance of the overall structure. Compared with AH-70 asphalt concrete, the long-term rutting resistance of the hard-grade asphalt concrete base structure was better.

### 3.3. Deflection

To further evaluate the bearing capacity of the hard-grade asphalt concrete base structure, we conducted biweekly deflection tests using an FWD (falling weight deflectometer)

during long-term performance observation. Figures 13 and 14 show the curves of the deflection at the central points and the areas of the deflection basin of the two pavement structures in the FWD deflection test from December 2016 to November 2021.

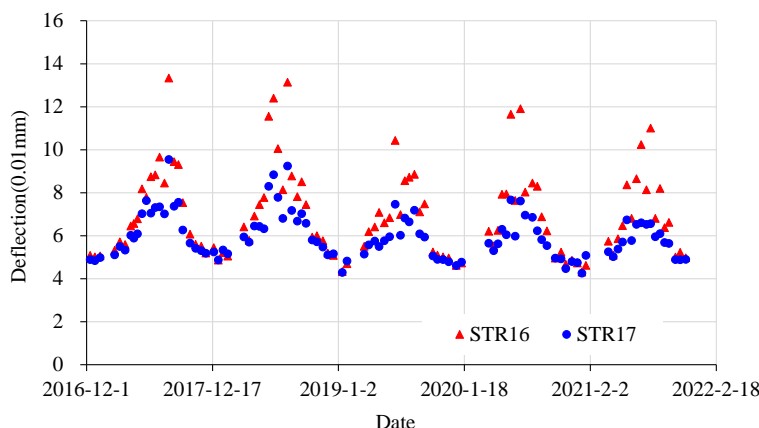

**Figure 13.** Evolution curve of deflection value.

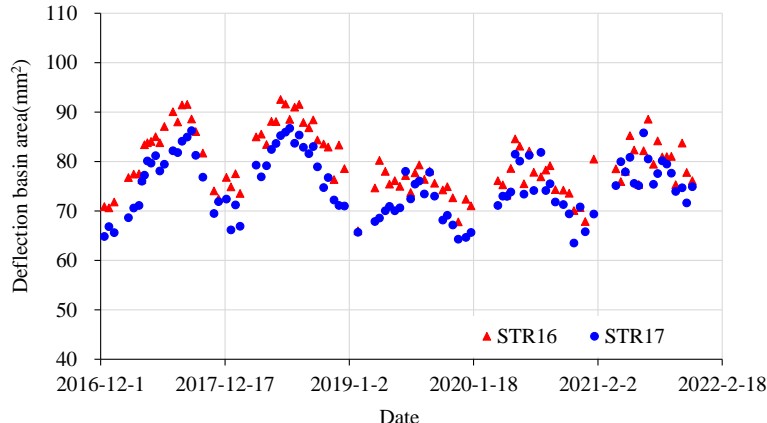

**Figure 14.** Evolution curve of deflection basin area.

The results in Figures 13 and 14 show that structure STR17, with a hard-grade asphalt concrete base, displayed smaller deflection at the central point, and a smaller deflection basin. According to the values in Figures 13 and 14, the differences in percentage between the deflection values at the center point and the areas of the deflection basins of two structures in the same test period were calculated, respectively. The results show that, compared with STR16, the center point deflection value of STR17 was 14.9% smaller on average, and the deflection basin area was 6.1% smaller on average, which indicates that STR17 had better structural bearing capacity. From the dynamic modulus test results presented in Table 4, we can see that, at 20 °C, the dynamic modulus of the upper base layer AC25 (AH-30) of STR17 was 19,189 MPa, while in the case of AC25 (AH-70) of STR16, it was 12,004 MPa. It can be seen that the dynamic modulus of the hard-grade asphalt concrete was nearly 60% higher than that of the AH-70 asphalt concrete at the same temperature. Due to the higher modulus, the pavement structure was more resistant to vertical deformation under loading.

To further analyze the influence of hard-grade asphalt concrete on the stress state of the pavement structure, the multi-layered elastic theory was adopted, the calculation software BISAR3.0 of Shell Company was used, and the parameters in Table 7 were used to simulate the single, circular, uniformly distributed load of FWD. The diameter of the loading circle was 30 cm, and the uniformly distributed load was 0.7 MPa. The modulus in Table 7 was determined through laboratory tests according to China's Specification for Design of Highway Asphalt Pavements (JTG D50-2017) [37]. The moduli of asphalt

mixtures AC25 (AH-30), AC25 (AH-70), AC20 (AH-30), AC20 (SBS) and SMA13 (SBS) are the dynamic moduli obtained by uniaxial dynamic compression tests. The conditions were 20 °C, and the loading frequency was 10 Hz, which are the same as in the tests discussed in Table 4. The moduli of cement-stabilized gravel (CBG25) and cement-stabilized soil (CS) are the elastic moduli obtained by uniaxial compression tests after the specimens were cured under standard conditions for 90 days. The modulus of soil is the resilient modulus of the top surface of the subgrade obtained after subgrade filling was completed during the RIOHTrack construction, and was tested with an on-site bearing plate test. In the area where the RIOHTrack is located, the undisturbed soil of the road surface is clay soil, with good physical and mechanical properties, so this soil is directly used for subgrade filling. The Poisson's ratio of each layer of pavement, shown in Table 7, is based on the requirements of China's Specification for Design of Highway Asphalt Pavements (JTG D50-2017), and no relevant tests have been carried out. Through the calculation of BISAR3.0, it was found that the vertical displacement value of the pavement surface of STR17 under single-circle uniform loading was 14.25 (0.01 mm), while for STR16 it was 15.46 (0.01 mm). Structure calculation was performed, showing that under single-circle uniform loading, the vertical displacement of the pavement surface of the hard-grade asphalt concrete structure was approximately 8.5% smaller than that of the AH-70 asphalt concrete structure, indicating that the former has better structural bearing capacity.

**Table 7.** Structural calculation parameters.

| | STR16 | | | | STR17 | | |
| --- | --- | --- | --- | --- | --- | --- | --- |
| Layer | Thickness (cm) | Modulus (MPa) | Poisson's Ratio | Layer | Thickness (cm) | Modulus (MPa) | Poisson's Ratio |
| SMA13 (SBS) | 4 | 7307 | 0.25 | SMA13 (SBS) | 4 | 7307 | 0.25 |
| AC20 (SBS) | 8 | 10,528 | 0.25 | AC20 (AH-30) | 8 | 17,486 | 0.25 |
| AC25 (AH-70) | 12 | 12,004 | 0.25 | AC25 (AH-30) | 12 | 19,189 | 0.25 |
| AC25 (AH-70) | 12 | 12,004 | 0.25 | AC25 (AH-30) | 12 | 19,189 | 0.25 |
| CBG25 | 20 | 11,265 | 0.25 | CBG25 | 20 | 11,265 | 0.25 |
| CS | 20 | 3256 | 0.25 | CS | 20 | 3256 | 0.25 |
| Soil | - | 58 | 0.4 | Soil | - | 58 | 0.4 |

## 4. Conclusions

The following major conclusions can be drawn from the long-term performance observations and research on the hard-grade asphalt concrete base structure:

(1) From the results of laboratory experiments, hard-grade asphalt concrete has higher strength, modulus, and high-temperature stability, but its low-temperature performance is generally inferior to that of AH-70 asphalt;

(2) It can be seen from the measured results of pavement cracking that in the area where the RIOHTrack is located, the AH-30 hard-grade asphalt concrete base structure underwent less transverse through-type cracking than the AH-70 asphalt concrete base structure. The total length of cracking was 3.4 m shorter, and the percentage of cracking was 14.8% less. The hard-grade asphalt concrete can basically guarantee a cracking resistance performance equivalent to or slightly better than that of AH-70 asphalt concrete;

(3) The long-term rutting curve shows that, compared with AH-70 asphalt concrete, the average rutting depth of the hard-grade asphalt concrete base structure was approximately 1 mm smaller, and the rutting resistance was about 16% higher, indicating the better long-term rutting resistance of the hard-grade asphalt concrete base pavement structure;

(4) During the five-year observation period, the phenomenon whereby the rutting of the pavement structure increased with annual fluctuations may have existed for a long time, but it can only be discovered when rutting detection is conducted more frequently, and the amount of rutting is recorded in all 12 months of the year. Understanding the cause of this phenomenon requires further study;

(5) The results of the long-term evolution of deflection show that, compared with AH-70 asphalt, the measured center point deflection value, deflection basin area, and theoretically calculated vertical displacement of the road surface of the hard-grade asphalt concrete base structure are 14.9%, 6.1%, and 8.5% smaller, respectively. This shows that the hard-grade asphalt concrete structure has a better structural bearing capacity.

This study shows that the hard-grade asphalt concrete base pavement is suitable for use in road constructions in areas of China with a climate similar to that of the RIOHTrack, or with a higher temperature or lower latitude.

**Author Contributions:** The authors contributed to the paper as follows: conceptualization and methodology: X.Z.; data curation: Y.W. and L.S.; formal analyzing and validation: Y.W., X.Z. and X.W.; writing and visualization: Y.W., X.Z. and L.S. All authors have read and agreed to the published version of the manuscript.

**Funding:** This work was funded by the National Key of Research and Development Plan under Grant number 2020YFA0714300.

**Institutional Review Board Statement:** Not applicable.

**Informed Consent Statement:** Not applicable.

**Data Availability Statement:** Not applicable.

**Conflicts of Interest:** The authors declare no conflict of interest.

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
