# Peer review of "Long-Term Service Performance of Hard-Grade Asphalt Concrete Base Pavement Based on Accelerated Loading Test of Full-Scale Structure"

_sustainability, doi:10.3390/su14159712_

Round 1

Reviewer 1 Report

This is a valuable research manuscript. Low-grade asphalt has better performance, but it is rarely used in China due to the lack of long-term engineering verification. The authors paved test roads based on the full-scale pavement structure to study the long-term service performance of the low-grade asphalt, which has theoretical and engineering value for the application of the low-grade asphalt. Some suggestions are listed below.

(1) The “low-grade asphalt” is misleading. The authors may consider using “hard-grade asphalt”.

(2) The expression in Figure 1 is not clear. It is suggested to modify it to highlight only the two structures, STR16 and STR17. The material properties of each pavement layer in Figure 2 should be given, such as the type of the asphalt binder, the particle size of the aggregate, etc., so that readers can understand the experimental conditions in detail.

(3) The units of standard axle load and truck axle load mentioned in the manuscript are both t. Equation 1 should be modified to use t as the unit. In the definition of Pi, the unit should be given.

(4) Please verify the 9.5mm sieve pass rate of AC20 (AH-30) and AC20 (SBS) in Table 2. Whether the gradation curves of AC25 (AH-30) and AC25 (AH-70), AC20 (AH-30) and AC20 (SBS) are the same, if they are the same, they should be combined and expressed. To be more intuitive, Table 2 should be supplemented with the corresponding gradation curve.

(5) In the research work on crack, the accumulative number of axle loads as of September 2021 should be stated. The type, parameters, layout, collection and test instructions of the temperature sensors embedded in the pavement structure should be given.

Reviewer 2 Report

Based on the accelerated loading test of full-scale pavement, this paper studies the long-term service performance of low-grade asphalt concrete base pavement through technical indicators such as cracks, rutting, and deflection, which can provide some reference and guidance for its popularization and use in China. Please see the following comments:

1. line 31: ‘…RIOHTrack…’, When abbreviations appear for the first time, please explain clearly, and the same is true in the text.

2. It is suggested that Part 3 and Part 4 be merged to make the structure of the paper clearer and to delete the repeated statements in the two parts.

3. It is recommended that Table 4 be shown in the form of a graph, and the analysis graph corresponding to Table 6 be added.

4. In Section 6.1, it is suggested to add actual crack photos to visually display the test results.

5. Please make the conclusions more concise.

6. English should be more polished before re-submission.

Reviewer 3 Report

GENERAL REMARKS

Introduction

The introduction is too general, insufficiently specific and has some inaccuracies:

- Reference [2] is not technical standard (lines 58-59);

- On what basis was the sentence "The hard asphalt with Pen10/25 or 15/25 is most used in Europe (2)" formulated? It does not result directly from the text presented in [2] (lines 47-48).

- Did the authors not mean the penetration grade of 10/20, mentioned in the work [2]?

The chapter should be expanded.

Objectives and scope

The content of the chapter should indicate specific data:

- What was the length of the test section (line 84)?

- how long did the tests take?

The phrase "in service for long enough" is insufficient (line 86).

The text will be much more readable if the authors move some of the text from the "Objectives and scope" chapter to "Introduction", and move the remaining significant part of the text to the next chapter.

In line 83, after the name of the Institute (Research Institute of Highway Ministry of Transport) should be added the abbreviated name of the track "(RIOHTrack)", which the authors use in the following chapters of the manuscript.

Methods

The authors wrote that the experimental road test section was tested for over five years. This is where the information on the test load should be provided (vehicle type and axle weight distribution, speed and frequency of passing of test vehicles, etc.). Then the authors presented some information about the road test section. In the next paragraph, the test track is described again and the test section load conditions are described in more detail ... Certainly, the text will be more consistent if the authors change the order of the 2nd and 3rd paragraphs.

Methods / General Situation … / Pavement material …

Further analysis of the text presented in the next two chapters shows that it is advisable to combine three chapters ("Methods", "General Situation ..." and "Pavement material ...") into one chapter (for example, "Materials and Methods"). It is also necessary to introduce 3 subsections, in which, without unnecessary repetitions, the authors will present detailed information on: full-scale road test track "RIOHTrack" (subsection 2.1), the structure and materials used to build the STR16 and STR17 test sections (2.2) and the methodology for implementing accelerated load tests (2.3).

Long-term service performance …

The authors should explain on what basis they formulated the following sentence: "Furthermore, the pavement structure STR17 using low-grade asphalt concrete as the base has better resistance to structural cracking than the pavement structure STR16 that uses AH-70 asphalt concrete as the base". Did the authors compare more than just the total length of the cracks?

At this point, it is worth adding at least information expressed as a percentage - what was the difference in the total length of cracks (14.8%).

Were samples taken at the locations of the cracks (inter alia, to determine the propagation / depth / width of the cracks)?

The statement “low-grade asphalt concrete… can guarantee a crack resistance equivalent to or better than that of the AH-70 asphalt concrete” is too firm (lines 216-217). The authors should not formulate such far-reaching conclusions in the case of the comparative analysis of the results obtained from only one test section (STR17) - the more so as in the case of the reference section (STR16 ), there were also shorter cracks.

Clarification is needed on how the difference of 70% was calculated (line 274).

The authors should also explain how they performed the calculations (analytically - or using computer software dedicated to this type of calculations). In both cases, additional information is needed to show how the results presented in Table 7 were obtained.

It is also necessary to explain on what basis for the calculations such and not other material parameters of the soil lying under the test track (stiffness modulus 60 MPa) were adopted. What exactly was the type of soil? If soil stiffness tests were performed, this should be written about.

Conclusions

The third paragraph is definitely too much emphasized - despite the fact that the comparative analysis (as noted earlier) is based solely on the comparison of the total length of the cracks.

There is not enough detail in the conclusions regarding the measurements carried out and the differences between the two test sections (by how many% are the total lengths of cracks different, what is the difference in deflections, etc.)

The phrase "... with a climate similar to that of RIOHTrack or with a higher temperature or lower latitude" appears twice in the Conclusions. This type of repetition should be avoided.

DETAILED COMMENTS

The names of the authors should be placed one after the other and separated with commas.

Affiliation should be prepared in accordance with the Template.

 1  Affiliation 1; e-mail@e-mail.com

2  Affiliation 2; e-mail@e-mail.com

* Correspondence: e-mail@e-mail.com; Tel.: (optional; include country code; if there are multiple corresponding authors, add author initials)

Abstract

The abstract contains 203 words. If possible, please shorten the text to a maximum of 200 words.

Keywords

Keywords should be written in lowercase (keyword 1; keyword 2; keyword 3)

The text of the manuscript

Main chapter titles except the first letter should be written in lowercase (e.g., Introduction, References).

Lines 44, 48, etc. References should be indicated by a numeral or numerals in square brackets—e.g., [1] or [2,5], or [3–6].

Line 47. The wording "Pen10 / 25" should be corrected to "penetration grade (PG) of 10 / ..."

Line 52. In the case of the word "Laboratory" you would have to use the plural ("laboratories").

No space (space) between consecutive sentences.

Lines 54-58. The entire sentence should be re-edited. It is too complex and difficult to read. In one sentence, the authors use the phrase: "the actual pavement structure" twice.

Line 58. No space before the parenthesis.

Lines 65-72. Too complex sentences. They should be redrafted. Authors are misusing the word "although".

Lines 77-78. The sentence needs to be redrafted. It is not fully legible / understandable.

Lines 105-108. Sentences should be rewritten (what was "AH-30 low-grade asphalt" used for?; replace "as of the end" with "by the end"; instead of "in" use "in line / in accordance with ...", etc.).

Line 118. Length units are usually separated by a space from the numeric value. Better change the value "2,039 m" to "2.039 km".

Lines 143-145. Move the sentence "According to ..." to line 150.

Line 143. At the end of a sentence, indicate the standard or other reference source from which the equation comes.

Lines 147-149. Equal signs "=" should be changed to "-".

Lines 162-163. Do not use Capitalise Each Word in table titles.

Line 163 (Table 2). Unify all numeric values expressed in% to 1 decimal place.

Line 176. Replace the word "crack" with "cracking".

Move Table 5 to line 193.

Table 6 should be moved to line 205.

In tables, units of numerical values are usually inserted after a decimal point or in square brackets. For example, in Table 1: "Ductility, cm" or "Ductility [cm]"; "Rotary viscosity, Pa · s" or "Rotary viscosity [Pa · s]".

Line 208. After the words "asphalt concrete" add "AC25 (AH-30)".

Lines 206-210. Re-edit the sentence that is too complex and delete the repetition of "the ultimate failure strain" (use the phrase "while / whereas in the case of").

Move Figure 6 to line 225.

Lines 228-229. Add the information by how many% differ the corresponding values from Table 4.

Line 237. Cross out "in the atmosphere".

Transfer Figure 7 to line 237.

Line 267. Expand the abbreviation FWD (falling weight deflectometer)

Move figures 8 & 9 to line 271.

Line 273. Correct to "results presented in".

On lines 282 and 284 references to the literature should be added.

Lines 281-293. Rewrite the sentence.

Author Contributions

Instead of full names and surnames, use initials.

References

The list of references should be corrected in accordance with the guidelines (among others, first indicate the surnames, and only then include the initials of the names).

Some items should be italicized (Abbreviated Journal Name, Volume, Book Title, etc.). In the case of Journals, the year of publication should be in bold.

Round 2

Reviewer 3 Report

The authors' responses are satisfactory.

The revised manuscript looks much better. It is more logical and clear.

Unfortunately, from the stylistic point of view, the text still requires some corrections.

 Both American English (A.E./US) and British English (B.E./UK) are used in the text.

Among others, in line 397 there is 'analyze' (US), while in line 456 there is the word 'analysis' (UK).

American English vocabulary is also found in lines 42 ('popularized'), 215 ('characterize'), 285 ('analyzes'), 362 ('characterize'), etc. Authors should consistently use American English or British English spellings.

Some of the references need to be improved.

Among others, in line 46 ('Spain [30]'), most likely the numbering should be changed from '[30]' to '[10]'. Consequently, subsequent references should also be renumbered.

Line 47. 'South Africa [13], Australia [13]' - should be changed to 'South Africa and Australia [13]'.

On lines 65 and 495, the author's surname and first name should be corrected ('Maria' - this is a given name, not a surname).

Reference [16] in line 76 ('Lee H J et al. [16]') should also be corrected.

In the captions to Figures, the punctuation marks should be corrected in accordance with the 'template' (Figures 4, 10).

Some of the sentences require redrafting. Among others in lines: 91-93.

Line 289 and others. Remove the space between the numerical value and the '100°C' temperature unit.
